# XplAInable: Explainable AI Smoke Detection at the Edge

Alexander Lehnert [1,*,†] , Falko Gawantka [2,†] , Jonas During [3] , Franz Just [2] and Marc Reichenbach [1]

1    Institute of Applied Microelectronics and Computation Engineering, Faculty of Computer Science and Electrical Engineering, University of Rostock, 18051 Rostock, Germany; marc.reichenbach@uni-rostock.de
2    Department of Computer Science, Faculty of Electrical Engineering and Computer Science, Hochschule Zittau/Görlitz, 02763 Zittau, Germany; falko.gawantka@hszg.de (F.G.); franz.just@hszg.de (F.J.)
3    Department of Computer Science, Brandenburg University of Technology Cottbus-Senftenberg, 03013 Cottbus, Germany; jonas.during@b-tu.de
*    Correspondence: alexander.lehnert@uni-rostock.de
†    These authors contributed equally to this work.

**Abstract:** Wild and forest fires pose a threat to forests and thereby, in extension, to wild life and humanity. Recent history shows an increase in devastating damages caused by fires. Traditional fire detection systems, such as video surveillance, fail in the early stages of a rural forest fire. Such systems would see the fire only when the damage is immense. Novel low-power smoke detection units based on gas sensors can detect smoke fumes in the early development stages of fires. The required proximity is only achieved using a distributed network of sensors interconnected via 5G. In the context of battery-powered sensor nodes, energy efficiency becomes a key metric. Using AI classification combined with XAI enables improved confidence regarding measurements. In this work, we present both a low-power gas sensor for smoke detection and a system elaboration regarding energy-efficient communication schemes and XAI-based evaluation. We show that leveraging edge processing in a smart way combined with buffered data samples in a 5G communication network yields optimal energy efficiency and rating results.

**Keywords:** edge computing; sensor network; machine learning pipeline; explainable AI; energy efficiency





## 1. Introduction

The automation of systems and the increasing digitization of processes lead to progress in many domains [1]. The nationwide implementation of 5G networks enables the realization of intelligent systems on a large scale [2]. Smart cities are conceivable applications of distributed and intelligent systems in urban areas, relying on large-scale communication networks. However, it is important to note that such intelligent systems are not limited to urban settings and can also be applied in rural areas [3,4].

The expansion of 5G as a high-performance communication medium allows for re-thinking conventional application architectures. Instead, very large application scenarios connect a multitude of sensors, actuators, and processing nodes. These novel concepts surpass the state-of-the-art implementations in terms of energy efficiency, performance, latency, and bandwidth [5–7]. The 5G standard foresees three distinct usage profiles, optimizing network coverage toward either low latency, high bandwidth, or a large number of devices in the network [8] (p. 112). While targeting single-use cases suffices for many 5G applications, projects requiring high performance in all regards are hindered by this principle. Communication becomes expensive, necessitating computation to be pushed toward the edge.

Wildfires pose a significant threat to flora and fauna [9]. For instance, between 2001 and 2021, forest areas equivalent to the size of Great Britain were burned worldwide [10]. From 2017 to 2022, the number of international large wildfire disasters ranged between 9

and 19 [11]. Therefore, wildfire prevention has become a critical task. The rise of artificial intelligence (AI) motivates the automation of surveillance and early detection systems. Modern fire detection systems have become cheaper, more reliable, and can be deployed even in highly rural areas. We present a project that addresses wildfire prevention by integrating sensor data into a distributed system, including cameras, stationary ground sensors for environmental conditions, novel smoke detection devices, and drone surveillance. In this work, we introduce a novel smoke detection system based on the Bosch BME688 gas sensor, augmented with edge classification and the simulation of data transmission costs, alongside an additional explainable AI (XAI) evaluation using a remote server.

As already alluded to, our case study consists of a smoke gas detection sensor based on the Bosch BME688 sensor. The evaluation of data samples recorded by this sensor are classified as smoke-induced air and fresh air. Communication is considered expensive; therefore, system-level models of energy costs for various communication schemes have to be explored. Additionally, in the context of wildfire prevention, a dispatcher is faced with the task of combining all sensor data recorded in the system to make a decision on setting an alarm for wildfires. This decision can be aided by the additional analysis of the recorded data samples. We explored explainable AI (XAI) methods of analyzing the raw data samples at a remote server location.

In this paper, we present the following contributions.First of all, this work explores and answered the following two research questions:

- RQ1: How can smoke detection be shifted to the edge of large-scale communication networks when classifying data samples from the Bosch BME688 sensor?
- RQ2: What benefits does XAI offer in the development and live evaluation of artificial intelligence for edge devices?

Further contributions of the work presented here are as follows:

- An MLP-based classifier for smoke gas deployed and optimized for execution using an ESP32-platform.
- A holistic system simulation of energy costs for various communication schemes from the sensor node via BLE.
- A design space evaluation in terms of the energy intensity of various communication schemes compared to the value of information received by a central server.

The rest of this paper is organized as follows. In Section 2, a case study is shown in a real-world use case. Section 3 provides an in-depth investigation of smoke detection on edge devices. Additionally, the benefits and uses of XAI evaluation are described in Section 4. Further, a comparison of different setups of on-the-edge hardware is described in Section 5, as well as how XAI is integrated in the verification step in the AI development pipeline. Finally, the results of our system-level evaluation are shown in Section 6, and alongside a conclusion, an outlook into further research is presented in Section 7.

## 2. Case Study—Project 5G-Waldwächter

Forest areas are severely threatened by climate change, and there are additional risk factors that further increase this threat [12]. Forest fires [10–12] and bark beetle [13,14] infestations of forest areas are severe threats to the local flora and fauna. Within the scope of the project 5G-Waldwächter, researchers and partners from the industry collaborated to present a holistic forest health monitoring and early alert system (5G-Waldwächter is a 5G pilot project of the district of Görlitz, Germany, funded by the German Federal Ministry of Digitalization and Transport (BMDV), focusing on the early detection of forest hazards [15]). In this context, we present the conceived distributed, intelligent detection system.

### 2.1. Main Idea of 5G-Waldwächter (Context for This Work)

The overall 5G-Waldwächter system uses camera data input, available data from public weather services, and specific data provided by other public services like EFFIS [16]. All different input data, i.e., camera systems, available weather services, and services for

wildfires, are processed and grouped into subsystems. After considering all information and validating each subsystem, a reliable prediction will be made.

One of these subsystems is a network of sensor nodes that can detect smoke. Detecting smoke is very important for two major reasons. The first reason is that a wildfire starts with smoke, so instead of detecting a fire with flames upon trees, smoke would be detected much earlier, and therefore, the ascending fire could be prevented through smaller activities. The second reason is that if there is dry grass and dead wood on the ground, a fire could burn with slightly visible information. So, this means that there is time between when the fire starts, its spreading, and the visual recognition above the peak of the trees. This is why monitoring the ground of the forest could provide crucial information about a very early stage of the fire.

However, technically, it is easier to observe big areas visually with cameras, but then the wildfire is harder to prevent (or to extinguish). The detection of an early stage is, from a technological viewpoint, harder because there are many open questions such as: "How many sensors do I need for 1 km$^2$?" But from the viewpoint of the actions necessary to fight the fire, the effort required is much lower. This is the reason why investigating such an approach is very fundamental for the early detection of wildfires. Here, a subsystem that can detect smoke comes into play; this part of the system will be covered in detail in the following section.

### 2.2. Subsystem for Smoke Detection on the Edge (Scope of This Work)

To provide details about the smoke detection part of the 5G-Waldwächter system, we will firstly provide an overview of the topology of this subsystem. This topology is shown in Figure 1. There are three major components to identify: the sensor node, the communication node, and the central server. Furthermore, the sensor node consists of a Bosch BME688 sensor, an ESP32, and one BLE module. The ESP32 MCU interfaces with the Bosch sensor and the BLE module. With the BLE module, the sensor node can communicate with the communication node. The communication node then interacts with the central server.

The ESP32 uses internal buses (SPI, I2C) for communication. Bluetooth Low Energy is used between a sensor node and a communication node. And the top-level communication of the system from the communication node to the central server is conducted via 5G.

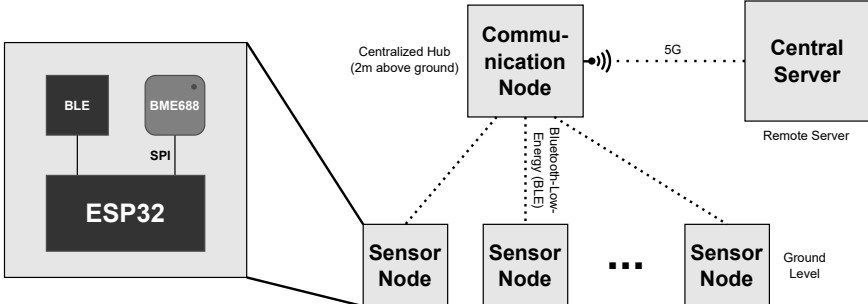

**Figure 1.** In this figure, the context of the edge sensor (i.e., sensor node), is shown. The communication node, as a transmitter and collector, is in the middle between the nodes and the Central Server.

After showing the components and their interfaces, the functions are explained in detail. The target of the distributed sensor nodes is to record data samples of various types, such as temperature, the humidity of the ground and the surrounding air, and permittivity and air pressure. Early preprocessing with the ESP (i.e., on the sensor node) could be accomplished by leveraging AI for the classification of the recorded data samples. This is performed at defined intervals, e.g., once every hour; the data can be stored or sent to the communication node. The functionality of the communication node is to collect data from multiple sensor nodes using Bluetooth Low Energy (BLE) over a short distance. The data are then buffered in this communication node for a certain period of time and later

transmitted to a remote server component (i.e., the central server) over a 5G connection. The proposed communication scheme above can be modified for special requirements.

The reasons (i.e., goals of the system) for adapting the scheme and the entities are presented in the following. First, all sensor nodes have resource costs, i.e., processing time and energy, associated with both measuring data, processing data, and transmitting the information to the next communication node. The longevity of the individual nodes is a key driver of the design decisions in conceiving the distributed node setup. For optimal detection, sensor nodes should be placed as close to the source of smoke particles as possible (i.e., at ground level). However, as the nodes are placed in the forest in a distributed manner, we cannot supply them with continuous energy (i.e., they are battery-powered). In terms of transmission errors from the sensor nodes to the communication nodes, a mechanism must be implemented (e.g., a buffer). A similar mechanism must be implemented for sending data from a communication node via 5G. All the entities themselves should react in terms of temporal connection issues.

These uncertainties, with their key premises on energy consumption, lead to the idea to slightly modify the system components in Figure 1, to address some of the issues that can arise, and to compare these system variations with each other. Therefore, in this case study, several operating options are presented and discussed in greater detail. An overview of the following options is shown in Table 1.

**Table 1.** Overview over various possible operating profiles of the processing and communication of the edge sensor node. The following abbreviations are used in the table: M for measuring, E for evaluation, S for sending, and class for classification. ✓ denotes support, while × denotes the lack of such.

| Costs | Possible Operating Profiles | | | |
|:---:|:---:|:---:|:---:|:---:|
| | **Data Sample Only** | **Class Only, No Sample** | **Class and Sample** | **Class and Multiple Samples** |
| M | ✓ | ✓ | ✓ | ✓ |
| E | × | ✓ | (✓) | (✓) |
| S | 1 sample | <1 sample (only class) | >1 sample (class + sample) | >$n$ samples (class + $n$ samples) |

The range of operation modes ranges from transmitting only the raw data samples to sending an anomaly classification tag and a specified number of buffered samples before the anomalous data point occurs. The choice in communication profile directly impacts energy consumption for processing and sending. The influence of all three aspects combined is estimated based on the costs of the profile that only transmits raw data. In conclusion, sending the raw data is the minimal use case that can be conducted. Further impacts of various other communication schemes on the overall system have to be explored. Therefore, we present an evaluation in terms of both energy and entropy, i.e., the value of information present at the Centralized Server. In this regard, the following aspects are explored:

- The quality of prediction or classification using AI at the sensor nodes;
- The quality of the overall smoke detection;
- Energy aspects of both the sensor node and the overall system;
- The feasibility of transmitting data over 5G in rural areas from many devices, possibly at high bandwidths;
- System reliability.

### 2.3. Constraints and Assumptions

The most important aspect of this system is energy consumption, which necessitates the use of low-power hardware, with the ESP32 being mandatory. In a broader context, 5G was selected for the 5G-Waldwächter system as the communication network. This decision was made because, for observing rural areas, using the newest generation of radio network is the most practical way of installing such a distributed system. Additionally, 5G supports

the concept called mMTC (massive Machine-Type Communication), which is a feature for interacting with a large number of devices (1 million devices per square kilometer), which could be necessary for observing the ground in terms of wildfires. However, as a communication alternative, 4G can be used as a fallback mechanism.

The reason for integrating explainable AI (XAI) into the AI validation process is to improve the machine learning model. With XAI, it is possible to deploy a more efficient model on the ESP32, enabling better predictions or requiring less power. Integrating XAI tools in this use case could provide valuable insights and lead to new communication schemes or usage profiles of the sensor nodes.

To be explicit, the current work is a proof of concept, with a strong focus on architecture and component design. Features related to privacy and security are considered on a basic level; however, they will not be the primary focus of this smoke detection system.

Our assumptions are that the BLE connection is highly stable and that the 5G radio network works properly in most cases. When making assumptions on advertising and technical limitations, we generally make worst-case assumptions, meaning that in most cases, the real system will perform better.

In summary, key points from this section include the value of detecting smoke with sensors at an early fire stage, the placement of smoke sensors on the ground, and their battery-powered nature. Therefore, the main driver for the system is to develop a reliable, energy-efficient, distributed, and intelligent system that detects smoke.

## 3. Smoke Gas Detection

### 3.1. Related Work

During the last decade, applications around electrical noses (e-noses) based on gas sensors [17] have gained popularity. This trend is enabled by the accuracy and low power requirements of state-of-the-art gas sensors. Among many applications, e-noses play an important role in air-quality monitoring [18–20] and smoke detection systems [21–24].

Among different types of gas sensors, such as electro-chemical gas sensors [25], infrared gas sensors [26], and optical gas sensors [27], especially metal oxide semi-conductor (MOS) gas-sensing devices have found use. While MOS gas sensors are a low-cost option, they provide low sensing delays and little sensor degradation compared to other sensor types [27]. MOS sensors are divided into non-resistive devices based on diodes, FETs and capacitors [28], and resistive sensors. Gas sensing based on resistance is achieved either through changes in surface properties induced by exposure to gases, or through the reduced gas adsorption of the metal surface of the MOS leading to low valency and, thus, changes in conductivity [29].

The Bosch BME688 [30] is a volume-controlled MOS gas-sensing device [31]. It is widely used in several applications [32,33] thanks to its low cost and accurate classification capabilities. The evaluation of the raw sensor data recorded by the BME688 is based on sensor fusion [34]. During a measurement cycle, the sensor performs a sequence of heating steps while simultaneously providing measurements for the temperature, gas humidity, pressure, and resistance of the metal surface at different steps of the cycle.

Energy-efficient communication is essential for low-power distributed sensors. Next to Zigbee, LoRa, and ultra-narrow-band devices, Bluetooth Low Energy (BLE) is a prominent communication standard especially for edge and internet of things applications [35]. Out of non-proprietary communication standards, BLE stands out as requiring very little power [36]. Sensor applications include proximity-based access control [37], motion tracking [38], noise monitoring [39], and medical sensors [40], to only name a few examples. Tosi presented an extensive study of the BLE performance [41]. While data rates of BLE are often lower in practice than in theory, they generally enable low-power communication. Meanwhile, systematic investigations of energy consumption for a given application and communication setup are required for proper energy evaluatios. According to Tosi, this aspect in particular is lacking in the literature.

In this work, we present a system-level approach to an edge evaluation application for the Bosch BME688 for smoke detection. The BME688 performs well at detecting smoke gas. Ref. [34] presented an artificial intelligence-based evaluation model. trained using random hyperparameter search that achieved a classification prediction of 99%. Further, Ref. [20] presented an air-pollution measurement to detect traces of wildfires. The approach presented here, in turn, involves a systematic evaluation of edge processing used to evaluate the BME688 measurements in the context of XAI for the detection of smoke particles, a corresponding confidence rating from an XAI evaluation, and the long-term assembly of datasets for future improvements.

### 3.2. System Model

With the target of detecting smoke gas outlined, we now go into the detail of the sensor setup. As already alluded to by other works, the Bosch BME688 is a volume-controlled MOS sensor [20,23,34]. One BME688 consists of a total of four sensors for temperature, humidity, pressure, and resistance, i.e., the actual gas detection. Measurements are performed in heating cycles [30]. During a heating cycle, the internal metal plate of the BME688 is heated to a sequence of defined temperatures for defined periods of time. At multiple points in time during one heating cycle, measurements of all four sensors are taken. The set of recorded values represents one data sample. The heating profile used in this work is presented in Figure 2a. In total, it consists of ten measurements, resulting in a data sample consisting of 40 values.

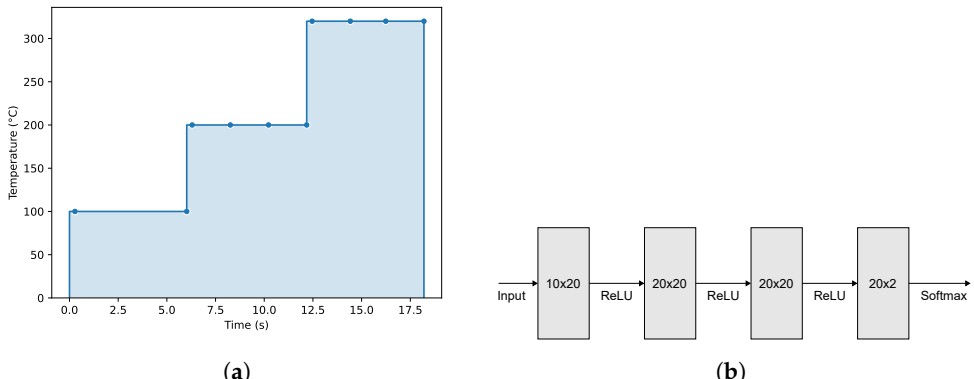

(**a**)  (**b**)

**Figure 2.** MLP and heating profile. (**a**) Heating profile. (**b**) Multi-Layer Perceptron used for classification.

Our goal was the evaluation of data samples on the edge, i.e., before transmitting the sample to a remote server. For this, we trained a simple Multi-Layer Perceptron (MLP), shown in Figure 2b, for classification into categories of smoke induced air and fresh air. The inputs of the MLP correspond to the sensor reading during the measurement profile shown in Figure 2a. With the data samples, we focus exclusively on the resistances among the 40 measured values, as they impact classification the most. This reduces the amount of data to ten values, thus resulting in less processing time. To mitigate the uncertainty in these measurements, we utilize batch normalization techniques during preprocessing. Following the normalization, the data are propagated through three hidden layers consisting of 20 neurons each. The final probabilistic classification is achieved with two neurons in the output layer.

The dataset used for training was recorded at a controlled wildfire. This ensures consistency in the environmental conditions between all measurements and that the resulting model can detect smoke not only under favorable conditions. Note that due to the circumstances of the dataset creation and a low number of recorded samples, we achieved a comparably low classification accuracy compared to other works [34].

As stated before, we intended to classify data samples at the edge, i.e., calculate the neural network on an ESP32-equipped FeatherS3 microcontroller board. The BME688 was connected to the introduced board, as shown in Figure 1. This allowed for the use of the

ESP32 SIMD instruction set extension for the efficient execution of the inference task in the trained model. Compared to a naive execution, i.e., without using SIMD instructions, we achieved an improvement of 55% in the execution time. All weights of the MLP were stored as C-macros, allowing for optimizations at compile time.

One key metric of a smoke gas detection system is the delay of detection. In the presented implementation, there are two factors, which together determine this delay. First of all, the pure detection period limits the detection delay. As indicated by the heating profile of the sensor (Figure 2a), one measurement takes 18 s. Compared to this, the classification and data transmission times are negligible. With some time spent establishing the BLE connection itself, the detection delay is less then 20 s. The second factor, which impacts the responsiveness of the system, is the measurement frequency. In our models, we assumed a frequency of one measurement each hour.

This concludes the recording and classification of data samples. Communication to the remote server is handled over BLE and 5G, as alluded to before. The communication node structure, in our case, is given by the overall setup described in Section 2. Energy-efficient execution and communication are the core goals. For this purpose, various communication schemes were modeled, as discussed in Section 5.

## 4. XAI

At this juncture, explainable artificial intelligence (XAI) was introduced as a field of study within the domain of artificial intelligence, accompanied by an explanation of its associated focuses and terminology. While traditional classification yields a labeling of data samples, the motivation for said labeling typically remains unclear. In the context of recorded sensor data, this issue hinders confidence in the correctness of said labeling. XAI achieves explainability of the classification and, thus, can aide decision processes.

Minh et al. [42] (p. 3511) defined XAI in their survey article as "the study of explainability and transparency for socio-technical systems, including AI". The 2019 XAI Taxonomy by Arya et al. [43] introduced essential terms in explainable artificial intelligence. Subsequently, Liao et al. [44] utilized this taxonomy as a decision-making tool to select appropriate explanatory algorithms for the IBM AIX 360 tool. The tree structure of Liao et al.'s taxonomy was transformed into a tabular format and can be observed in Figure 3.

This section aims to provide a brief overview of explanatory models applicable to the proposed use case. The table excludes the data leaf, and there is currently no information in the interactive path. Model explainability takes both a local and a global path, i.e., it distinguishes between explaining a data sample and explaining the predictive model. Local interpretations of a single data instance can be conducted ante hoc, where the predictor is comprehensible enough that a closer look serves as an explanation, for example, in the case of a simple decision tree (ML model).

On the other hand, the XAI post hoc methods require at least one execution of the predictor to provide information about the decision process of the ML model. The global model explanation is divided into two parts: direct methods, which involve direct interpretable notions and could be considered ante hoc, and post hoc methods, which generate explanations after executing the predictor.

| *model* | | | | | |
|---|---|---|---|---|---|
| *local* | | | *global* | | |
| ante-hoc | *post-hoc* | | direct | *post-hoc* | |
| | samples | features | | surrogate | visualize |

**Figure 3.** Modified algorithm overview based on the IBM XAI Taxonomy. It was derived from the decision tree presented in [44] and is a guide to the selection of an appropriate XAI method. This overview specifically emphasizes the model explanation aspect and does not present specific algorithms.

For the prediction problem proposed in the case study, a variety of XAI approaches capable of explaining any classifier would aid in providing objective explanations for predictions. According to the described taxonomy, post hoc algorithms that explain each feature of an instance could improve our understanding of the model's decision-making process. A selection of potential explainer methods is outlined in Table 2.

**Table 2.** Three distinct operational XAI algorithms are introduced with the aim of gaining insights into the decision-making process behind the predictions. These algorithms meet specified constraints, and their conceptual approaches, along with their sources, are provided.

| XAI Method | Constraints | Approach (Background) | Sources |
|---|---|---|---|
| LIME (Section 4.1) | | creates a surrogate model (functional approximation) | [45,46] |
| SHAP (Section 4.2) | fulfilled | shapely values are approximated (game theory) | [46,47] |
| CIU (Section 4.3) | | perturbed instances are used for Multiple Criteria Decision Making (MCDM) | [46,48] |

### 4.1. LIME in Detail

LIME operates by treating the local model as a black-box entity. The approach involves introducing perturbations to an initial data point, feeding it into the black-box model, and utilizing the resulting predictions to train an interpretable surrogate model. This surrogate model serves as a local approximation to the predictions made by the black-box model. The explanation generated by LIME is determined by

$$\xi(x) = \operatorname*{argmin}_{g \in G} \mathcal{L}(f, g, \pi_x) + \Omega(g) \tag{1}$$

In Equation (1), $\xi$ represents the explanation for the instance $x$, obtained through an optimization task. The function $g$ denotes an interpretable local model chosen from the class $G$ of potentially interpretable models. The function $f$ serves as the original predictor, and $\pi_x$ defines a sampling distribution concentrated on the local neighborhood of instance $x$. The optimization task seeks the optimal interpretable local model $g$ by minimizing a composite term. This term includes the loss function $\mathcal{L}$, which gauges the accuracy of predictions around $x$, concerning both the interpretable model $g$ and the original prediction of $f$ within the area of $\pi_x$ around the initial prediction. Additionally, $\Omega$ acts as a complexity measure for $g$ and serves as a penalty function [45].

### 4.2. SHAP in Detail

The foundation of SHAP lies in game theory, specifically in assessing the contribution or non-contribution of a coalition (a set of features) to a particular classification using Shapley values. This approach is based on the implementation introduced by Lundberg and Lee [47]. This implementation, referred to as SHap Additive exPlanations, calculates the Shapley values to quantify the influence of each feature in the coalition. The subsequent definition outlines how the algorithm generates explanations [47,49]:

$$g(z') = \phi_0 + \sum_{i=1}^{M} \phi_i z_i' \tag{2}$$

In Equation (2), the function $g$ represents the explanatory model, and $z'$ denotes the data instance to be interpreted. Notably, $z'$ may consist of only a subset of all features. The explanation is derived from a linear model, where $\phi_i \in \mathbb{R}$, and $z_i'$ takes binary values (either zero or one) to signify the presence or absence of a feature from the set $z'$ [47].

*4.3. CIU in Detail*

The third model-agnostic algorithm, influenced by Decision Theory and Multiple Criteria Decision Making (MCDM), separates measured importance and attribute utility, emphasizing contextual importance (*CI*) based on feature relevance [48,50].

$$CI_j(\vec{C}, \{i\}) = \frac{C_{max_j}(\vec{C}, \{i\}) - C_{min_j}(\vec{C}, \{i\})}{absmax_j - absmin_j} \tag{3}$$

The explanation model CIU calculates the relevance of feature $i$ in the feature vector $\vec{C}$ for a specific output label (value) $j$ using the function $CI_j$. The function $C_{max_j}$ determines the maximum prediction output for label $j$ considering the presence of feature $i$, while a similar approach is followed for the calculation of $C_{min_j}$. The functions $absmax_j$ and $absmin_j$ establish the highest and lowest prediction outputs from the entire dataset, providing a reference range.

In essence, $CI_j$ normalizes the range of predictions for label $j$ in the context of feature $i$ within $\vec{C}$ by comparing it to the overall range of predictions across all contexts and feature sets. For a more comprehensive understanding, refer to the paper by Kary Främling [48,50,51].

**5. Idea and Methodology**

In conclusion for the previous section, XAI achieves the desired explainability of sample classification. To make use of this, data samples have to be processed at a Centralized Server. The processing of XAI algorithms at the edge is infeasible due to the corresponding high energy costs. The following section discusses the modeling of the energy and power requirements of the sensor node. The simulation is based on various evaluation and data emission strategies targeted at achieving the desired transmission of samples to the Centralized Server and leverages physical measurements of power consumption. The physical sensing and network structure itself are presented in Section 3. Data are collected with the sensor on the ESP32-equipped microcontroller board. Communication over BLE requires advertising to establish a connection, as well as the actual data transmission. The evaluation of the data can be performed either on the edge, i.e., using the sensor-equipped microcontroller itself, or with the server.

There are two methods of lowering the cost of communication: fewer communication cycles or a reduced payload of transmitted data. One communication cycle, as alluded to, consists of both advertising and data transmission. In general, having fewer communication cycles, i.e., buffering data, cuts down on the cost of advertising, but not on the actual transmission of data. The preprocessing of data before transmission achieves the opposite effect: the number of communication cycles remains constant, but the payload of each cycle is reduced. Further, joint communication schemes can save even more energy.

We modeled the presented set of communication schemes by calculating the energy required to execute each one. Therein, each individual step, such as a measurement of the sensor or transmission of data are inferred from the physical power measurements of the ESP32-platform. This procedure provides an accurate energy estimation for each profile and thus enables a holistic system overview. Theoretical power draws of the individual phases of execution as well as actual measurements are presented in Table 3. Note that the literature provides little information on the power draw during advertisement. Further, the time required for successfully establishing the BLE connection, i.e., advertisement, varies a lot. In our model, we approximate the time with 0.5 s. By choosing a relatively small advertising delay, we ensure a pessimistic modeling of the communication. In practice, larger delays increase the cost per communication cycle. Therefore, the proposed preprocessing and communication schemes are even more relevant compared to the presented baseline.

**Table 3.** Overview over theoretical (if applicable) and measured power draw of the ESP32. For BLE advertisement (denoted by), no clear information is given in the literature.

| Phase | Theory | Measured | |
|---|---|---|---|
| | Power | Power | Time |
| Measurement | 8 µA | 1.7 mA | 18.2 s |
| Evaluation | 72.4 mA | 168 mA | 47 µs |
| Advertisement | * | 116 mA | 0.5 s |
| Transmission | $\frac{L}{275\,\mathrm{B\,s^{-1}}}$ | 116 mA | $\frac{L}{275\,\mathrm{B\,s^{-1}}}$ |

In the following, first, various steps of the system execution will be discussed in detail. Afterward, we explain our simulation model in detail, which provides results for the design space of evaluation and transmission configurations.

*5.1. Execution Steps*

5.1.1. Measurement

Each data sample is recorded using the Bosch BME688 sensor connected to the microcontroller using the heating profile presented in Figure 2a. During the measurement step, the ESP32 does not perform any additional computations. As the duration of the measurement cycle is known in advance, deep sleep is used to save energy. According to the specification of the ESP32 [52], we expect a power draw of 10 µA. Our measurements indicate a power draw of 1.7 mA, which is due to the connected sensor and board periphery.

5.1.2. Evaluation

First of all, data have to be collected from the BME688 sensor. The BME setup is explained in Section 3. For each measurement cycle, ten measurements of temperature, pressure, humidity, and resistance are recorded. Optionally, each data sample is evaluated using the MLP presented in Section 3.2.

We achieve efficient execution for the MLP for evaluation by leveraging the ISE of the ESP32. The ESP32 supports SIMD via vector instructions, which our MLP implementation leverages. Due to the small size of the MLP, a single execution time cannot be measured. Therefore, we provide an average execution time, as well as a power draw over 20,000 sequential executions. In total, our measurements indicate a power draw of 168 mA over an average execution time of 47 µs per classification.

5.1.3. Transmission

As alluded to previously, data transmission from the ESP32-based platform is conducted via BLE. Communication consists of an advertising period followed by the transmission itself. As advertising delays are highly inconsistent, in our simulations, we conservatively approximated the delay by 0.5 s. The transmission time period itself depends on the transmitted data payload. Our setup is not representative of optimal (laboratory) environments; on average, we achieved a data rate of $275\,\mathrm{B\,s^{-1}}$.

Each sample of measured data consists of readings from four sensors over ten time steps, each value encoded as a single precision floating point. Therefore, one data sample consists of 160 B. Evaluated data, on the other hand, consist of the classification results only, i.e., 8 B of data. Note that exceeding the obvious options of sending raw data, in the classification, raw data, if any, can also be buffered and sent at a later time. We define the payload $L$ as the number of bytes to be transmitted in one communication cycle (Equation (4)). $L$ depends on buffered data samples $b$ and classification setting $c_c \in 0, 1$, which determines whether classification results are included in the transmission.

$$L = b \times 160\,\mathrm{B} + c_c \times 8\,\mathrm{B} \qquad (4)$$

In practice, even more communication setups are viable. From a system perspective, the relevance of the measured data sample depends on the classification result, i.e., when-

ever smoke is detected, the corresponding data samples should be available for further examination using the presented XAI methods at the central server. We extended the model of the data transmission load to the one presented in Equation (5) with the classification result constraint $c_r \Leftrightarrow 1$, if the classification indicates smoke, otherwise, 0.

$$L = c_r \times b \times 160\,\text{B} + c_c \times 8\,\text{B} \tag{5}$$

*5.2. System Model*

Based on the various behavior configurations of the system, a multitude of communication and evaluation schemes follow. The chosen metric for our models is the energy spent for the given device behavior. We present the holistic simulation used to explore this design space, which is created by the following constraints:

- The place of executing the evaluation;
- Transmit data samples or raw data;
- Buffer data samples.

Each of the below-presented modes represents one cycle of collecting and handling a measurement sample. Additionally, each model contains one sleep step. During this sleep step, the sensor node is inactive. In our model, each configuration sleeps for the same amount of time. This does not infringe on the validity of our modeling, as the time for recording and handling data is very short compared to the relatively long sleep periods.

5.2.1. Buffered Send Mode

The first configuration considers the naive sensor node, which does not perform evaluation on an edge but transmits every data sample. Further, samples can be buffered. This saves energy, as the number of advertising rounds for establishing the BLE connection are reduced. For a given sample buffer size $b$, Equation (6) shows the energy of the `buffered send mode`. The various stages are labeled as `sleep` ($S$), `measure` ($M$), `advertise` ($A$), and `transmission` ($T$). For means of readability, we introduce abbreviations for the given stages, i.e., $E_S$, $E_M$, $E_A$, and $E_T$ represent the energy of the `sleep`, `measure advertise`, and `transmission` stages, respectively.

$$E = \underbrace{\frac{UI_S}{t_S}}_{E_S} + \underbrace{\frac{UI_M}{t_M}}_{E_M} + \frac{1}{b}\underbrace{\frac{UI_A}{t_A}}_{E_A} + 160\,\text{B} \times \underbrace{f_T UI_T}_{E_T} \tag{6}$$

Note that for $b = 1$, the trivial sensor node behavior follows, which sends every recorded data sample without buffering.

5.2.2. Classification-Based Mode

As already alluded to, we expect the majority of data packages to be negative classifications, i.e., the detection of no smoke. In turn, data samples classified positively are relevant for further evaluation in the context of the overall system. XAI methods, such as the ones presented in Section 4, allow for access to further information on the data samples. Therefore, it is desirable to transmit these samples to the server, while the negative ones are disregarded. Communication behavior in this mode is strictly dependent on the classification outcome. Data samples are only transmitted if classified positively. Additionally, a window of recently buffered measurements $w$ is transmitted. The energy of the `classification mode` is given in Equation (7), where $C$ indicates the `classification` stage. Most notably, the energy depends on the probability $p$ of detecting a positive smoke sample.

$$E = E_S + E_M + \underbrace{\frac{UI_C}{t_C}}_{E_C} + p(E_A + w \times 160\,\text{B} \times E_T) \tag{7}$$

Note that for $w = 1$, this behavior reflects a configuration of sending positively classified samples only. Further, we chose a relatively high value of $p = \frac{20}{365}$ for our models. Similar to the estimation of the advertising period, this resembles a pessimistic assumption; as said, p would correlate with 20 days of forest fire each year.

### 5.2.3. Buffered Classification Mode

Previous modes discussed have been implemented with various ways of transmitting data samples. When evaluating data on the sensor node itself, this is not necessary though. The overall communication load is decreased, which leads to energy savings. In turn, further processing with XAI algorithms at the server becomes infeasible. For the classification mode, all classification results are sent immediately post evaluation, while the transmission of data samples is omitted entirely. The energy of this mode is given in Equation (8). As with the `buffered send mode`, energy can be saved by buffering results and advertising less.

$$E = E_S + E_M + \frac{1}{b}E_A + 8\,\text{B} \times E_T \tag{8}$$

### 5.2.4. Standalone Mode

Similar to the concept of the `classification-based` modes, the previously presented `classification mode` can also be extended to only emit positive classification results. While this behavior leads to further immediate energy savings, the functionality of the sensor node cannot be guaranteed. It can be assumed that the negative classifications are not only the norm, but also, the greatest part of the classifications turn out negative. Thus, differentiation between a dysfunctional sensor node and a functional sensor node becomes infeasible over long periods of time. We propose a rare but regular emission of negative classification results as a sign of life. Such a signal is sent with frequency $f_L$, and the corresponding energy is given by Equation (9):

$$E = E_S + E_M + (p + f_L)(E_A + 8\,\text{B} \times E_T) \tag{9}$$

### *5.3. Power Evaluation*

Based on the various modes presented in Section 5.2, we present an evaluation of several communication models. A general overview is given in Table 4.

The first category of communication modes are `send modes`. Data samples are transmitted (mostly) without prior classification at the edge. Key differences between the presented modes are the number of buffers used. Four buffer sizes are presented ranging from one to four. The naive send mode is equivalent to the buffered send mode with a buffer size of one. Additionally, classification can be performed by the sensor node, and results can also be transmitted. Said behavior is modeled by the `send mode`, including classification.

The second category of modes are `classification modes`. These modes leverage the edge processing of classification at the sensor node and, thus, can achieve lower power requirements. This is especially true for the probabilistic modes, as these transmit data only upon positive classification results, i.e., once smoke is detected. With the environmental dependency at play, we chose a comparatively high value for said probability at $p = \frac{20}{365}$. We ensured to model a worst case performance, as this probability value is equivalent to smoke being detected on 20 days in a single year. Further, data samples may be buffered and transmitted at once. Note that while pure send modes benefit from buffered samples, the same is not true for classification-based modes. With the increased payload per communication cycle by buffering samples, we observed an increase in the power requirements.

**Table 4.** Overview of energy, power, and capacity for various modes. The chosen probability is $p = \frac{20}{365}$.

| Name | Buffer | Energy (J) | Power (mW) | Capacity (As) |
|---|---|---|---|---|
| Send Mode (incl. Classification) | 1 | 23.754 | 299.4 | 7.198 |
| Send Mode | 1 | 23.752 | 235.6 | 7.197 |
| prob. Classification Mode | 1 | 23.337 | 154.6 | 7.197 |
| prob. Classification-Only Mode | 1 | 23.334 | 154.6 | 7.181 |
| Classification Buffer Mode | 2 | 23.616 | 261.1 | 7.214 |
| | 3 | 23.608 | 248.3 | 7.231 |
| | 4 | 23.632 | 242.0 | 7.248 |
| prob. Classification Buffer Mode | 1 | 23.337 | 154.6 | 7.197 |
| | 2 | 23.340 | 154.6 | 7.214 |
| | 3 | 23.343 | 154.6 | 7.231 |
| | 4 | 23.346 | 154.6 | 7.248 |
| Buffered Send Mode | 1 | 23.752 | 235.6 | 7.197 |
| | 2 | 23.560 | 187.8 | 7.197 |
| | 3 | 23.496 | 171.8 | 7.197 |
| | 4 | 23.464 | 163.8 | 7.197 |

Finally, there is a baseline requirement for any sensor node setup, i.e., the sensing using the BME688 sensor. This represents the bare minimum in terms of energy consumption. At this point, no data are transmitted to the server at all; thus, this is not a realistic scheme. The previously mentioned `standalone mode` requires slightly more energy than the probabilistic classification-only mode, as it emits sign-of-life signals at a regular interval. The energy requirements of all modes presented are compared in Figure 4. The lower and upper line represent the baseline, i.e., sensing only, and the naive send mode without any buffering, respectively. From this overview, the energy efficiency of the probabilistic modes can immediately be inferred. For reference, we equipped our sensor node with a 10 Ah battery. In this setup, and using the probabilistic classification buffer mode with a window of 1, the node stays alive for 5002 cycles, or just over 208 days. The next evaluation step is the XAI evaluation followed by the combination of all the results in Section 6.2.

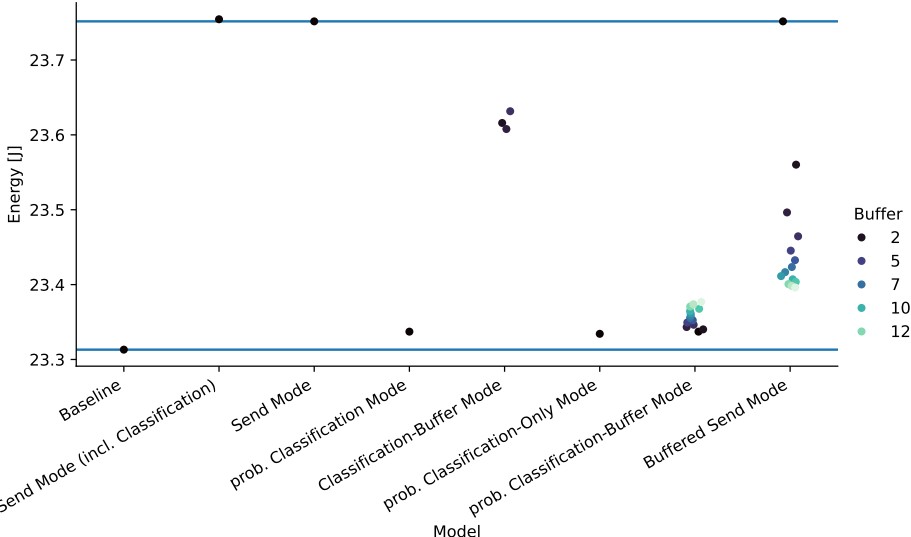

**Figure 4.** Comparison of energy requirements for models with various buffer sizes. The lines represent the minimum energy required for measuring and sleeping, and the base line of the naive model.

## 6. Results

This section begins by presenting the results of the XAI evaluation in a clear and intuitive manner. Subsequently, these results are contextualized to provide a comprehensive understanding of energy consumption, system performance, and optimization mechanisms tailored to the requirements of the 5G-Waldwächter project.

### 6.1. XAI Evaluation

As proposed in Section 4, integrating XAI into the development process can provide insights into the model's decision-making process. This information can be leveraged to streamline the model architecture, thereby reducing energy consumption, or to identify the best-performing predictors and, importantly, the most significant features. Explanation methods can also be employed to detect bias in the ML model.

Initially, the XAI algorithms were solely interpreted. For example, the results of the LIME algorithm are depicted in Figure 5. The figure illustrates the 25–75 quartile of the feature importance values, including the median, the remaining distributions, and any outliers.

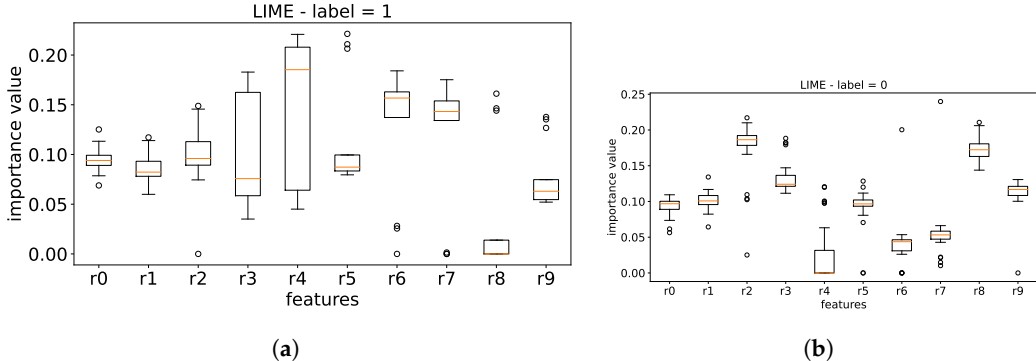

(**a**)                                         (**b**)

**Figure 5.** Comparison of of the explanation algorithm LIME for both labels. (**a**) Feature importance values for smoke detection. (**b**) FIV for detecting the absence of smoke.

According to RQ1, identifying the most relevant features for the classification process can significantly aid in fine-tuning the model. The median, denoted by a straight orange line within the boxplots in Figure 5a, serves as an initial indicator of the feature importance. For the top three most relevant features, the results are r4, r6, and r7 in descending order. To assess the distribution and evaluate result reliability, the spread of the values is considered. Among these three crucial features, the boxplot from LIME exhibits the widest spread. However, beyond the median up to Q3 (the third quartile), no other feature exerts as significant an influence as r4, indicating its paramount importance. Feature r6 demonstrates that values between the median and Q3 are tightly clustered, with the maximum value aligning with r7. Consequently, r6 may be considered more reliable above the median. In summary, the trustworthiness evaluation in LIME ranks r6 highest, followed by r7, and then r4. An overview of the LIME evaluation results is presented in Table 5.

**Table 5.** The results of the individual evaluations of the XAI methods are based on two aspects: the importance (i.e., relevance) and trustworthiness of the explanations generated by different explainers.

| Intra Algorithmic | LIME | | | SHAP | | | CIU | | |
|---|---|---|---|---|---|---|---|---|---|
| | **1st** | **2nd** | **3rd** | **1st** | **2nd** | **3rd** | **1st** | **2nd** | **3rd** |
| relevance$_{(trust)}$ | $r4_{3rd}$ | $r6_{1st}$ | $r7_{2nd}$ | $r4_{2nd}$ | $r3_{1st}$ | $r5_{3rd}$ | $r3_{1st}$ | $r2_{2nd}$ | $r4_{3rd}$ |

As a validation of the top three features for Label 1 (indicating smoke detected), there is no indication that the importance values of Label 0 (indicating no smoke) play a role in this analysis.

In the SHAP algorithm, as shown in Figure 6b, feature r4 emerged as the most prominent, followed by r3, and finally r5, completing the top three. Among these, r3 stands out as the most reliable feature. When considering values from the median to the maximum value, encompassing 50% of the data, r4 exhibits greater reliability than r5. However, if all values above the first quartile (Q1), where 75% of the data lie, are taken into account, r5 appears more trustworthy than r4. Nevertheless, since no other feature surpasses the median value of r4, and 50% of the crucial values are concentrated within a narrower range compared to r5, the confidence in r3 remains the highest, followed by r4, and then r5. This evaluation summary is presented in Table 5.

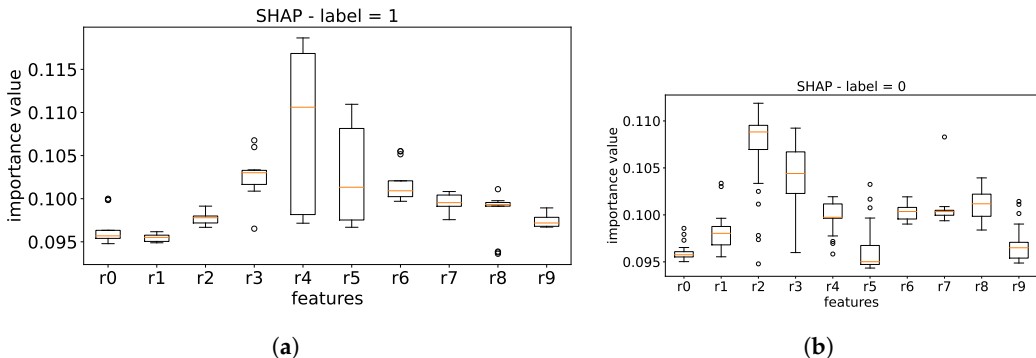

(**a**)  (**b**)

**Figure 6.** (**a**) Boxplot of the feature importance values of the SHAP algorithm for Label 1, i.e., for detecting smoke. (**b**) Boxplot of feature importance values of the SHAP algorithm for Label 0, i.e., for detecting the absence of smoke.

In comparison to Label 0 (indicating no smoke), feature r3 also exhibits significance, reinforcing its role in the decision-making process for both smoke and no-smoke scenarios.

The top three importance features of CIU are, as shown in Figure 7b, r3, r2, and, finally, r4 based on the median. The confidence of these top three, considering the upper 50% of the values, results in r3, r2, and finally r4.

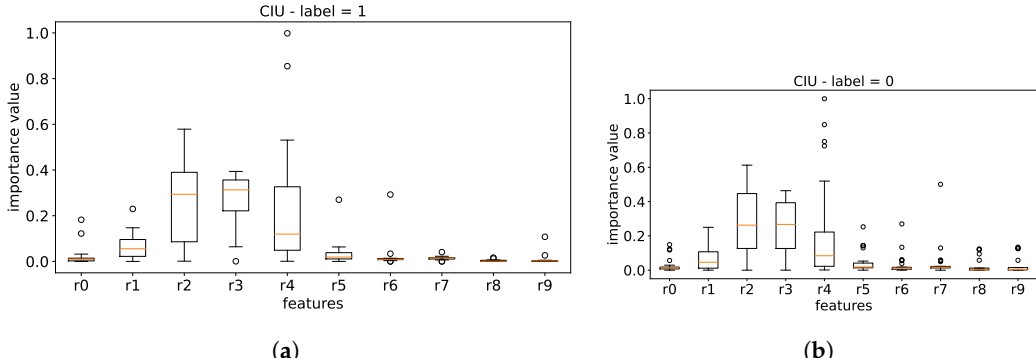

(**a**)  (**b**)

**Figure 7.** (**a**) Boxplot of feature importance values of the CIU algorithm for Label 1, i.e., for detecting smoke. (**b**) Boxplot of feature importance values of the CIU algorithm for Label 0, i.e., for detecting the absence of smoke.

Compared to Label 0, the same features play a crucial role in the prediction process without smoke. This supports the importance and the key role in the decision process for the CIU algorithm. An overview of these results is given in Table 5.

In the proposed use case, a framework of three different XAI algorithms with various calculation models was used. The result of applying these three algorithms to the predictor over the whole test dataset is shown in Figure 8.

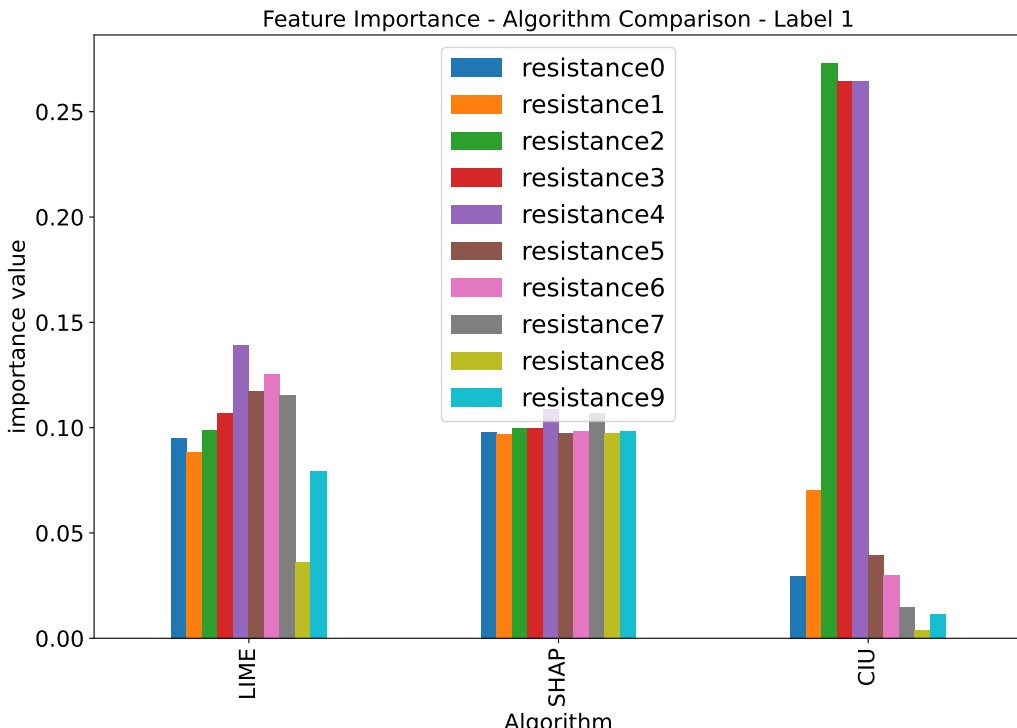

**Figure 8.** Here, a comparison of the three XAI methods is shown. The inputs for all XAI algorithms were the ML model and the test dataset. The comparison of the three should reveal the most important gas resistant features for detecting fire.

As shown in Figure 8, the gas resistance feature resistance4 (r4) has an outstanding impact across all three algorithms and has to be considered in the decision process. The feature importance value for feature resistance8 (r8) shows great variations. This means that it is not a relevant feature and should not be considered. The feature resistance3 (r3) is above average in LIME and SHAP, but in CIU, it shows great significance, suggesting that it is important. Compared to with LIME and SHAP, the feature resistance9 (r9) shows, only in SHAP, an average impact for classification. In LIME and especially in CIU, this significance is not present, and therefore, it should not be considered. Another feature that shows inferior performance is resistance1 (r1). A closer look reveals that it performs not well in LIME, but it has only an average importance in SHAP and a greater negative influence in CIU. Overall, this feature should not be considered. After this analysis, while peeking inside the black box, i.e., the predictor, the three features (resistance1 (r1), resistance8 (r8), and resistance9) should not be considered in an improved version of the ML model. A summary of this evaluation is presented in Table 6.

**Table 6.** Here, is a comparison of the feature importance values. Confidence in the explanations was neglected due to the clarity of the diagram. On the other hand, this comparative approach allows for the easy identification of less relevant features in the comparison, which is an aspect that is gained with this evaluation technique.

| | Combined Result | | | | |
|---|---|---|---|---|---|
| **Inter Algorithmic** | **Top** | | **Worst** | | |
| relevance$_\text{overall rank}$ | r4$_\text{1st}$ | r3$_\text{2nd}$ | r8 | r9 | r1 |

After evaluating the explanations using two different comparison approaches—within an algorithm and between algorithms—the results are summarized. The question "What are the most relevant features?" can be confidently answered with the finding that *r4* consistently plays a major role in every algorithm, as indicated in Table 5. This conclusion

is strongly supported with the comparison of 'inter-algorithmic relevance' in Table 6. Additionally, the significance of feature $r3$, highlighted in Table 5, is also evident in Table 6, further reinforcing the findings of the 'intra-algorithmic relevance' analysis.

The aspect of trustworthiness, examined within the 'intra-algorithmic relevance', can serve as an indicator of the certainty of an explanation. This aspect can be useful for fine-tuning the predictor and for interpreting the quality of the explanations—whether they are consistent or require careful consideration.

Another optimization option lies in the information about the least important features revealed in Table 6. These data can address optimization queries such as "Can we reduce computational steps by omitting certain features to save energy?". Clearly, there are compelling reasons to implement an eXplainable AI pipeline for identifying optimization approaches and assessing the effectiveness of optimization steps.

*6.2. Joint Evaluation*

In the previous sections, the experimental setup consisting of the sensor, the ESP32 for the evaluation, and the communication pipeline were presented. Furthermore, the design space of the communication schemes and the XAI evaluation was discussed. The following section presents an evaluation based on the previously discussed communication models jointly with the quality of the XAI evaluation. There are several metrics at play here. First and foremost, we evaluated the energy requirements of the sensor node. This is most important, as this edge node is battery dependent. Secondly, we rate the information available to the central server or database. Two use cases were identified for the application of XAI in the presented domain:

- Data credibility rating;
- Long-term improvement of datasets.

Firstly, the results from the XAI evaluation will be used as additional information provided to the dispatcher. Secondly, we use the positively classified samples in combination with negative samples underlying equal exterior conditions for a new holistic dataset.

The evaluation of information gained by the application of XAI algorithms is not straightforward. In turn, we define the gained information based on the two presented use cases. Classification credibility relies on the data sample being available at the server. Any communication model omitting the transmission of the samples themselves does not satisfy this requirement. To create new datasets in the long term, not only positively classified data samples have to be available, but also a window of corresponding negative data samples should be stored. Therefore, any model that is able to buffer samples achieves this second condition. The value of information degrades with the number of additional samples being available. We propose the following simple cost model for the value of the information $V_I$ based on buffered samples $b$, as presented in Equation (10):

$$V_I = \frac{1}{\log b + 2} \tag{10}$$

Based on this cost model and the power and energy evaluation presented in Section 5.3, Figure 9 presents an overview of all explored models.

A pareto plot (Figure 9 shows a clear winner for the communication schemes. The pareto frontier is dominated by the probabilistic classification-based mode with buffer windows. Further, the baseline and the standalone mode are part of the pareto frontier. Note that the baseline here does not reflect an actual communication scheme, as it considers the cost of recording samples only. The standalone mode, to no surprise, has the lowest communication overhead. Information, i.e., data samples, available at the central server location increases with the buffer window size. The same holds also for the buffered transmission mode without any evaluation, but it is more expensive in general. Again, this simulation is based on pessimistic assumptions about the advertising period and the probability of detecting smoke. In a real-world application, we can expect the probabilistic

classification based-mode with buffer windows to outperform the buffered transmission mode even more.

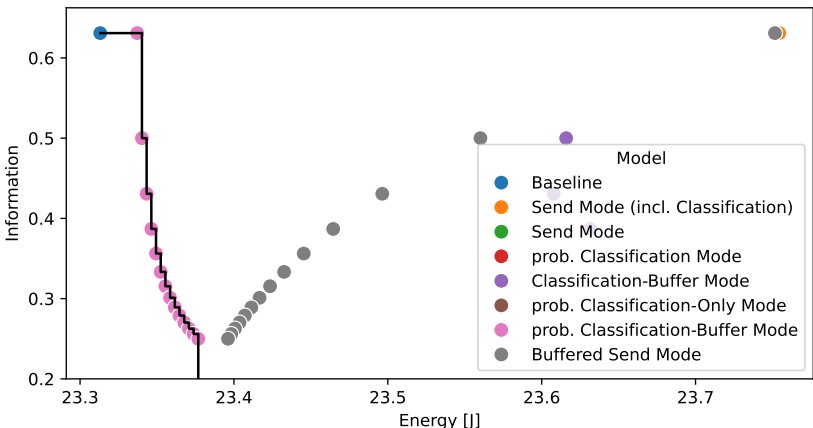

**Figure 9.** Pareto plot of energy cost and the value of information transmitted to the server. The information rating was calculated according to Equation (10). The black line marks the pareto frontier.

## 7. Conclusions and Future Work

The following section presents a conclusion of the presented work. Then, an outlook into future research directions extending our work is provided.

### 7.1. Conclusions

The underlying case study presents an intriguing use case for preventing large wildfires. We introduced a gas sensor, referred to as the sensor node, which serves as a smoke detector. This is coupled with an edge evaluation and a system-level energy simulation for various communication schemes directed toward a central server. Furthermore, we demonstrated the value of XAI methods in providing additional insights from recorded data samples, which is a key benefit of integrating XAI (answering RQ2).

To provide a comprehensive system overview, we presented two use cases for processing data samples centrally after evaluating at the edge. We utilize SIMD instructions to accelerate inference in MLPs. Firstly, data samples were used to create larger datasets that accurately reflect environmental conditions, independent of laboratory conditions. Secondly, by leveraging live XAI evaluations, we established a confidence rating for recorded data samples. This rating can serve as additional information in the overall system, guiding dispatching decisions or indicating impaired sensor nodes. The buffered transmission of data samples upon positive smoke classification poses the best trade-off as an energy-efficient communication strategy that also gives rise to centralized confidence ratings from an XAI evaluation.

The first research question (RQ1) can be answered affirmatively: it is possible to shift accurate predictions to the edge with the same level of confidence as processing at a central server. However, additional mechanisms need to be implemented when shifting calculations to the edge, particularly in handling communication errors, such as network downtime. One approach is to implement a buffer or integrate external storage for a period of time. Further research is needed to address such challenges. There is no exact sensing range forof our Bosch-BME688 based sensor node, as this capability depends on many environmental variables, such as terrain properties or wind conditions. In general, the smoke gas detection range of one sensor is in the range of 20 m. Our investigation relies on two key assumptions: firstly, the consistent functionality of the 5G network, and secondly, the efficiency of the MLP calculation. To ensure sustained system performance, implementing a cyclic adaptation process for the model would be beneficial for continuous improvement.

In summary of the second research question (RQ2), considering all aspects of integrating XAI into such a system, three primary benefits emerge. Firstly, it offers insights into the

model, enhancing our understanding of the detected values. The integration of an XAI tool chain for long-term model enhancement is feasible. Secondly, this additional information can aid dispatchers in discerning when to trust the system and when to exercise caution in their decisions. Lastly, XAI enables the detection of biases, particularly important when creating large datasets, thereby addressing issues arising from underrepresented data.

*7.2. Future Work*

From here, there are several aspects to explore. In this paper, we show that individual features provide confidence scores of MLP classifications. The same XAI evaluation can provide even more desirable information. In combining multiple features through different XAI algorithms, a more sophisticated confidence score can be generated. In addition, XAI can provide long-term support for the MLP design process. In reducing the computational accuracy of unimportant features and changing the network structure, the overall cost of evaluating data samples can be reduced without compromising the quality of the classification. The same applies to the measurement step. Optimized heating profiles can be implemented by comparing the effects of time steps in the measurement cycle on classification with the heating periods and temperatures of the measurement.

As already alluded to, a long term study of the active sensor and communication system is required to evaluate the benefit of applying XAI to improve datasets. Our approach to XAI yields confidence scores for recorded data samples. This additional information is useful for balancing datasets pre-training. In this way, long-term data collection can be leveraged to improve the MLP classifier of the smoke gas sensor node.

A further shortcoming of the presented work is the evaluation with regard to the stability of communication. All presented simulations were based on the assumption of functional communication without the loss or corruption of data. In reality, this condition is not always met. Further work on the presented energy simulations should regard the instability of communication as an additional factor.

Many more aspects can be elaborated upon in future work. In the presented work, we did not account for the distance of the gas sources to the sensor. Further, environmental effects, such the composition of the environment and effects from the wind or sun, can impact the data samples and, thus, the quality of the classification. Finally, the security of the communication of classification results and raw data samples is open for future investigation.

All sources will be made public in the future. This includes the set of data samples that the Bosch BME688 collected at a controlled wild-fire scenario, as well as further data collected at the experiment, such as temperature, humidity and images from nearby firewatch towers. Further, the simulation framework, as well as the neural network and the mapping framework, will be made public.

**Author Contributions:** Conceptualization, A.L., F.G. and M.R.; Methodology, A.L.; Software, J.D. and F.J.; Validation, J.D.; Formal analysis, A.L.; Investigation, F.G., J.D. and F.J.; Data curation, J.D. and F.J.; Writing—original draft, A.L. and F.G.; Writing—review & editing, A.L., F.G., J.D. and M.R.; Visualization, F.G. and F.J.; Supervision, A.L., F.G. and M.R.; Project administration, M.R.; Funding acquisition, M.R. All authors have read and agreed to the published version of the manuscript.

**Funding:** This research was funded by the German Federal Ministry of Digitalization and Transport (BMDV) under grant number 45FGU108_D.

**Data Availability Statement:** The data used in this work is part of a holistic data set collection consiting of smoke-gas sensor readings, environmental sensor readings such as temperature, air pressure and humidity of the air and the ground, image data (pictures, rgb, acromatic, near-infrared, temperature) and video footage captured by drones. The authors plan on making the holistic dataset of a wild-fire publicly available in the future. At this point, the legal processes for publishing the data have not been finalized yet.

**Conflicts of Interest:** The authors declare no conflicts of interest.

**Abbreviations**

The following abbreviations are used in this manuscript:

| | |
|------|------|
| AI | Artificial Intelligence; |
| XAI | Explainable Artificial Intelligence; |
| FIV | Feature Importance Value; |
| MLP | Multi-Level Perceptron; |
| SIMD | Single-Instruction, Multiple-Data; |
| BLE | Bluetooth Low Energy; |
| MOS | Metal Oxide Semi-Conductor. |

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
