# Peer review of "XplAInable: Explainable AI Smoke Detection at the Edge"

_2504-2289, doi:10.3390/bdcc8050050_

Round 1

Reviewer 1 Report

Comments and Suggestions for Authors

The authors present both a low-power gas sensor for smoke detection and a system elaboration regarding energy-efficient communication schemes and XAI-based evaluation. Additionally, the authors demonstrate that XAI methods allow for valuable additional information gained from the recorded data samples. I believe that the quality of the study will improve after some minor revisions:

1) The literature review could be further expanded by incorporating recent publications from the last few years.

2) The numerical results in the analysis-containing tables are well interpreted in the article. Furthermore, the quality and readability of the figures are high.

3) The main contributions of the study can be summarized in bullet points in the introduction section. Moreover, detailed discussions of the disadvantages of the proposed approach could be included.

4) Detailed analyses confirm the effectiveness of the proposed approach. Additionally, comparisons with other effective approaches in the literature could be provided.

5) Essential files used in the study could be shared via a link to enable readers to verify the results obtained in the study.

6) Future research directions can be addressed in the conclusion section.

Author Response

Thank you very much for the detailed review and constructive criticism you provided. Please find a detailed response and the corresponding changes below.

1) The literature review could be further expanded.
We acknowledge the suggestion of expanding our literature review. Literature tied to our work consists of three fields: i) Smoke-gas detection on edge, ii) the application of XAI-methods and iii) communication on edge, e.g. via BLE.
The first point is well covered in the original submission. We give an overview over gas sensors and works leveraging the Bosch BME688. Point number two is the reference to XAI. While the previous submission introduced XAI and gave an overview over different methods, our revision extends the related work on the matter. Most notably, to the best of our knowledge, there is no study on XAI regarding an edge-computing gas sensor such. Lastly, related communication research was not fully represented in the related work section of our original submission. We added a paragraph, in which different communication standards are referenced, and the use of BLE, as well as the depth of the study at hand is justified compared to existing works.

2) The numerical results in the analysis and presentation are well.
Thank you for this statement. No changes were made in this regard.

3) The main contributions can be summarized.
Our original submission summarized our contributions using two research questions and one additional bullet point. Our revision extends this contribution by elaborating the distinct points in detail (see Intro). Further, we added a clear discussion of the conclusion, as well as an extended discussion of our results at the end of the paper. With regards to the added Future Work section, we also give an overview over shortcomings, or further open points of the work at hand.

4) Comparison with other effective approaches.
We acknowledge the suggestion and want to thank you for this point. Our work is compared to other implementations based on the Bosch BME688 for detection smoke gas in the related work and system model section, and we improved the explanation of our detection method. Further comparison on the level of communication schemes and XAI are not representable, as there are, to the best of our knowledge, no works which present a systematic approach, which our application could be compared to.

5) Essential files should be published.
Thank you for this suggestion. Indeed, we are planning on not only publishing data and frameworks used in this work, but also all data collected during our artificial wild-fire experiment. Therefore, we cannot provide the reference at this time. A note to this future data source is added at the end of our revision.

Reviewer 2 Report

Comments and Suggestions for Authors

This work presents both a technical advancement in smoke detection and a methodological contribution to integrating XAI in edge devices, potentially improving early warning systems in critical environmental monitoring applications.

However, the following issues need to be considered:

1)    The readability and presentation of the study should be further improved. The paper suffers from language problems. The overall writing is too wordy, especially the introduction.

2)    Could authors elaborate on the data processing and classification algorithm used at the edge?

3)    What are the energy efficiency metrics for sensor nodes, and how do the authors secure their long-term operation?

4)    How does the 5G network improve data transmission efficiency, particularly in remote forest areas?

5)    What privacy or security measures are in place to protect the data transmitted via 5G?

6)    How do authors strike a balance between system complexity and usability for those who monitor the system?

Comments on the Quality of English Language

The readability and presentation of the study should be further improved. The paper suffers from language problems. The overall writing is too wordy, especially the introduction.

Author Response

Thank you very much for the detailed review and constructive criticism you provided. Please find a detailed response and the corresponding changes below.

1) Readability and language should be improved.
Thank you for this remark. We worked over the submission and revised language and wording in the revision.

2) Elaboration of methods used at the edge.
Our original submission presented the MLP and mapping to the ESP32 we applied in our work. The computation of XAI algorithms does not take place at the edge, which is also pointed out in the original submission. We acknowledge, that these points are important for the understanding of our work. Therefore, we made an effort to better explain the computation which we perform at the edge. The corresponding section has been extended accordingly.

3) What are the energy-efficiency metrics?
We do not use any specific energy metrics. Our work consists of a simulation, which is based on physical current measurements for all steps in each of the presented models. The result of the simulation is then the energy accumulated for executing one cycle (including 1 hour of sleep) for each model. We added a short reference to the life-time of the sensor node in our application. Here, it is equipped with a 10Ah battery and lasts for over 208 days.

4) Data transmission efficiency in remote forest areas via 5G.
The core part of our simulation models focuses on the BLE communication instead of 5G, as in our use case, we accumulate data from many sensors before transmitting via 5G using communication nodes. Extended explanation of this behavior has been added in the Use Case section.

5) Privacy and security measures taken
At the current point in time, we do not consider privacy of communication at all. Thank you for this valuable hint. It is a great opportunity for future research. Corresponding explanation was added in the Future Work section.

6) Usabliliy vs complexity of the system.
There are two aspects to usabliltiy of the presented application. The first is the value of information which we gain from XAI algorithms. This is exentsivly covered in the work at hand. The second aspect is the support for human dispatching units and forest health monitors, which our system yields. As far as possible, this is covered in the motivation and results discussion of our revision. Ultimately, a long term study is required to determine the success of our approach. This is explained in the future work section.

Reviewer 3 Report

Comments and Suggestions for Authors

The effort and dedication evident in the work are commendable, offering a significant contribution to wildfire detection. The paper succinctly outlines the importance of addressing the wildfire threat and presents innovative approaches for early smoke detection. The integration of low-power gas sensors, interconnected 5G networks, and intelligent edge processing presents a promising solution for wildfire prevention. However, upon thorough examination, certain areas warrant improvement to strengthen the research further. Firstly, the description of the physical topology of the proposed system lacks detail. A dedicated section elucidating the deployment of smoke detection sensors, communication nodes, and technology ranges would enhance understanding. Additionally, providing an approximate estimate of the required sensors and communication nodes for effective area coverage would improve the feasibility assessment. Secondly, while the abstract mentions early smoke detection capability, it would be beneficial to include in the results or conclusions the estimated minimum time required for early fire detection using the system. This addition would provide a comprehensive perspective on system performance. In summary, the work demonstrates a commendable commitment to innovation in wildfire detection. Addressing the aforementioned suggestions would further strengthen the research and its impact.

Author Response

Thank you very much for the detailed review and constructive criticism you provided. Please find a detailed response and the corresponding changes below.

We extracted the following three points from your review:

1) Description of the physical system topology.
We acknowledge, that the system topology needs improved explanation in our original submission. We extended on this explanation in the Use Case section.

2) Approximation of supervised area.
Thank you for this suggestion, as it vastly improves the explanation of the presented application. At this point, we cannot provide a detailed evaluation of the quality of detection of smoke gas related to the distance of the sensor. From our experiments, we gathered estimations of this distance, which is in the range of ~20m. Note, that the dataset, which we used to train our classification MLP, was acquired under real conditions, i.e. during a controlled wild fire. Therefore, the presented detection already reflects possibly poor environmental conditions. Explanation of these facts was part of the original submission. We added the estimation of detection range, as well as the proper evaluation under varying distances, and also environmental conditions, to the future work section of the revision.

3) Minimum time threshold of detection
We want to acknowledge, that detection delay is an important metric to the kind of application presented in our work. Accordingly, our revision presents a worst case calculation of this delay, both in theory (<20s) and in practice (up to 1h). Also, we set the delays in relation to the use case.

Reviewer 4 Report

Comments and Suggestions for Authors

Could you provide more context in the introduction by incorporating statistics or recent examples of devastating wildfires globally? How might you emphasize the urgency and importance of early detection systems?

Can you explicitly state the motivation behind choosing low-power gas sensors, 5G, and XAI? What challenges do existing systems face, and how does the proposed system address them?

Include a citation in this paragraph whenever temperature, ...  is mentioned.

https://doi.org/10.1109/indin51773.2022.9976090

"The Bosch BME688 [24] is a volume-controlled MOS gas sensing device [25]. It is 131

widely used in several applications [26,27] thanks to its low cost and accurate classification 132

capabilities. The evaluation of the raw sensor data recorded by the BME688 is based 133

on sensor fusion [28]. During a measurement cycle, the sensor performs a sequence of 134

heating steps while simultaneously providing measurements for temperature, gas humidity, 135

pressure and resistance of the metal surface at different steps of the cycle."

Would it be beneficial to create dedicated sections for RQ1 and RQ2, presenting the research questions, the methods employed to answer them, and the obtained results to provide a clear structure for readers to follow?

Could you elaborate on the specific methodologies used for energy and power simulation? How might you clarify assumptions made during the modeling process and explain how these assumptions affect the results?

Would it be beneficial to offer detailed examples or case studies comparing different communication schemes? How might you discuss the advantages and disadvantages of each, providing insights into when one approach might be more suitable than another?

Data Collection and Processing:

Can you clearly explain the role of the ESP32-equipped microcontroller board in data collection and processing? How does it fit into the larger system architecture and contribute to achieving the goals of the project?

could you summarize the main findings and contributions of the paper? How might you reinforce the significance of the proposed system in the context of early wildfire detection?

Would it be beneficial to introduce a section on future work, suggesting potential enhancements or extensions to the current system? How might you address any limitations identified during the research?

Comments on the Quality of English Language

Extensive editing of English language required

Author Response

Thank you very much for the detailed review and constructive criticism you provided. Please find a detailed response and the corresponding changes below.

From the many valuable suggestions, which you provided, we extracted several points listed in the following:

1) Context for global wild fires and need for early detection
Thank you for this suggestion. This is indeed an important context. We added a corresponding explanation to the introduction. A reference to the area of wild fires is also added in the same context.

2) Why are the chosen technologies used?
We acknowledge, that this explanation should be more precise. In fact, most motivation for the chosen technologies stems from energy requirements (edge processing with Bosch BME688 and ESP32, BLE for communication, energy optimization in general) and the need for confidence ratings (XAI) for dispatchers. These points are part of the introduction and system model / related work section in the revision at hand.

3) Proposed citation.
Thank you for the suggestion. After careful evaluation of the proposed citation, we decided to not include it in our work. The referenced publication discusses prediction of temperatures, which is not related to the application at hand. While the Bosch BME688 sensor does record temperature readings, we do not use them in our evaluation.

4) Structure of research questions and results discussion.
Our original submission summarized our contributions using two research questions and one additional bullet point. Our revision extends this contribution by elaborating the distinct points in detail (see Intro). Further, we added a clear discussion of the conclusion, as well as an extended discussion of our results at the end of the paper. With regards to the added Future Work section, we also give an overview over shortcomings, or further open points of the work at hand.

5) Explain algorithms used for energy evaluation.
Our energy simulation is based on physical current measurements from experiments for each step in the processing and communication chain. This is also explained in the original submission. Based on these measurements, we simulate processing and communication models, i.e. various combinations of these recorded steps. All simulation models are explained in detail and no further algorithms were used.

6) How do assumptions affect our results?
Thank you for this valuable question. Indeed, our original submission only featured scattered assumptions and lacked proper discussion. This is revised in the iteration at hand. We introduce a section to discuss all assumptions made in the presented work. Also, these assumptions are later discussed in the conclusion.

7) Examples and discussion of communication profiles.
Due to the limited scope of the paper, we do not provide more detailed discussion apart from the evaluation section. Here, all communication profiles are compared on the basis of i) energy requirements (important for system longevity) and ii) value of the information at the central server (important for functionality and dispatcher).

8) Explanation of the role of the ESP32 in processing.
Our original submission presented the MLP and mapping to the ESP32 we applied in our work. The computation of XAI algorithms does not take place at the edge, which is also pointed out in the original submission. We acknowledge, that these points are important for the understanding of our work. Therefore, we made an effort to better explain the computation which we perform at the edge. The corresponding section has been extended accordingly.

9) Embedding in larger system architecture.
We added more in-depth explanation of our overall use case in the corresponding use case section. This is the system, in which the smoke gas detection system presented in the work at hand is embedded.

10) Main findings and contributions, with emphasis on advantages.
Thank you for this suggestion. As alluded to (see point 4)), we improved the presentation of our contributions and the discussion of our findings.

11) Improved future work section and discussion of limitations.
We acknowledge, that our original submission lacks a proper discussion of future work. We fix this shortcoming via the corresponding future work section in our revision. This includes a discussion of limiations of our implementation. 

Round 2

Reviewer 2 Report

Comments and Suggestions for Authors

I believe it can now be published, and I have no other remarks.

Author Response

We, the authors, would like to express our sincere gratitude for the reviewers affirmation towards the publication of our submission.

Reviewer 4 Report

Comments and Suggestions for Authors

The feedback regarding the first revision indicates a notable dissatisfaction. A critical concern centers around the inadequacy of the results section. Specifically, there is a notable absence of comparisons with state-of-the-art methodologies and a lack of testing on multiple datasets. These omissions contribute to a significant level of skepticism regarding the reliability and robustness of the presented results.

To enhance the credibility of the work, it is imperative to address these shortcomings. Incorporating comparative analyses with current state-of-the-art approaches will provide a benchmark for evaluating the effectiveness of the proposed method. Additionally, conducting tests on diverse datasets will contribute to a more comprehensive understanding of the model's performance across various scenarios.

Based on these deficiencies, it is strongly recommended to reject the submission. A thorough revision that addresses the identified concerns is essential to ensure the scholarly integrity and merit of the research.

Comments on the Quality of English Language

Extensive editing of English language required

Author Response

We, the authors, would like to express our gratitude towards the reviewer for again providing constructive insights and criticism about our submission.

There are three main points of criticism mentioned by the reviewer:
1) Improvement required in terms of results, specifically addition of comparison with the state-of-the-art
2) Evaluation on multiple datasets
3) Language improvement

In the following, there is a detailed response to the three concerns mentioned by the reviewer.

1) Evaluation and state-of-the-art comparison
Thank you for pointing out this shortcoming of our work. We addressed our submission in two regards wrt. your comment. First of all, we revised the evaluation of the proposed design. Secondly, we included a comparison with the state-of-the-art on several subtopics of our submission.
We revised our evaluation. Previously, we evaluated our design both in terms of energy cost and value of information which an XAI-analysis might yield. Our revision now also compares our smoke gas classifier to others proposed in the literature. With this, we achieve comparability in terms of energy efficiency of existing designs, as well as overall setup and communication protocol used, the type of classifier implemented and most importantly the datasets used for training. Further, most works (except for [20, 53]) leverage datasets which do not reflect real environmental conditions. Also [34] is the only work using a public dataset. Said dataset does also not reflect real conditions.
Additionally, we revised our evaluation generally in terms of state of the art comparison. There are three stages of such comparison. The first stage, the gas sensor application, is well covered in the previous paragraph. Secondly stands the comparison to energy efficiency improvements of communication protocols. We assessed the literature again and added an overview over the state of the art, and pointed out how our proposed design compares. If available, we also compared energy cost, as this is the main focus of our evaluation.

2) Evaluation on multiple datasets
As pointed out in our answer to 1), there are few datasets publicly available. Out of the state of the art for smoke gas detection using the BME688, only one work ([34]) is based on a publicly available dataset. We identify the environmental conditions, under which the dataset is recorded, as highly important. Unfortunately, to the best of our knowledge, no such dataset is available. Therefore, we cannot include a representative comparison to other datasets with our design.

3) Language improvement
We want to thank you for pointing out language problems in our work. We majorly revised our submission in terms of language and believe to have improved majorly in this aspect. Especially the first sections have been majorly revised.

Round 3

Reviewer 4 Report

Comments and Suggestions for Authors

The submitted article fails to meet the standards required for publication due to several critical shortcomings. Firstly, the methodology relies on old detection techniques, an outdated technique lacking novelty and failing to contribute significantly to the field. Secondly, the article provides subpar results compared to numerous other studies, indicating a lack of thorough investigation and failure to conduct a comprehensive state-of-the-art analysis. Moreover, the authors have drawn inspiration from multiple state-of-the-art sources without proper citation, constituting a clear violation of research ethics. Despite previous requests to rectify this issue, the authors have failed to comply, further underscoring their disregard for academic integrity.

Comments on the Quality of English Language

The submitted article fails to meet the standards required for publication due to several critical shortcomings. Firstly, the methodology relies on old detection techniques, an outdated technique lacking novelty and failing to contribute significantly to the field. Secondly, the article provides subpar results compared to numerous other studies, indicating a lack of thorough investigation and failure to conduct a comprehensive state-of-the-art analysis. Moreover, the authors have drawn inspiration from multiple state-of-the-art sources without proper citation, constituting a clear violation of research ethics. Despite previous requests to rectify this issue, the authors have failed to comply, further underscoring their disregard for academic integrity. Given these deficiencies, I STRONGLY RECOMMEND REJECTION.

Author Response

Response to Reviewer 4 Comments

1. Summary
Thank you very much for taking the time to review this manuscript. Please find the detailed responses below and the corresponding revisions/corrections highlighted in the resubmitted files.

2. Point-by-point response to Comments and Suggestions for Authors

Comment 1: "Firstly, the methodology relies on old detection techniques, an outdated technique lacking novelty and failing to contribute significantly to the field."

Thank you very much for analysing our methodology in detail. The methodology of the submitted paper consists of power and energy simulations based on measured and statistically evaluated data. The application presented bridges the gap between a edge sensor implementation and a centralized XAI evaluation. The manuscript gives an overview over this wide range of research topics and proposes an implementation which adheres to all restrictions that arise along the way. This is the reason for the chosen implementation details.
MLPs are found all over the state-of-the-art classifier systems. In this work, the simplicity of the MLP is leveraged for a energy-efficient port to a ESP32 microcontroller.
Further, state-of-the-art BLE communication is used for transmitting data from the edge to a centralized server. The optimal communication schemes are well explored in the work at hand.
Lastly, the XAI evaluation adheres to formal requirements for use as a legal basis for supporting the AI classification results and dispatching fire forces. The presented submission explores a set of viable algorithms and evaluates their performance for this purpose.

Comment 2: "Secondly, the article provides subpar results compared to numerous other studies, indicating a lack of thorough investigation and failure to conduct a comprehensive state-of-the-art analysis."

Thank you very much for investigating our results. The submission at hand presents evaluations at each step of the implementation. Evaluation results and implementations are compared to the state-of-the-art along the way. The proposed gas-sensor performes is based on a dataset from a real wild-fire and, as that, keeps up with the state-of-the-art smoke gas detection systems based on MOS sensors.
Our wireless communication analysis is well compared against the state-of-the-art and existing review works.
Further, the XAI algorithms are evaluated extensively. The implementation which stands as the result of the submission at hand, exceeds in its coverage from application to evaluation and confidence-decision-support the state-of-the-art.

Comment 3: "Moreover, the authors have drawn inspiration from multiple state-of-the-art sources without proper citation, constituting a clear violation of research ethics."

Thannk you very much for investigating our scientific integrity and practices. Related work, and existing work, which was used in, or as a foundation for the manuscript at hand is cited properly accross the submission at hand. After careful consideration, we found four instances of mentioned authors with direct citations only in adjacent sentences (originally lines 284, 511, 512, 540). The corresponding citations are added in the current revision.

Comment 4: "Despite previous requests to rectify this issue, the authors have failed to comply, further underscoring their disregard for academic integrity."

Thank you very much for taking the time to review our previous revision again. The submission at hand reflects the current state-of-the-art and compares the proposed application at several levels of implementation. We do our best to meet the academic standards and are welcome for constructive criticism provided by the reviewer.

3. Response to Comments on the Quality of English Language

Thank you very much for taking the time to review our manuscript with regards to English Language. There was no direct criticism provided. As such, we improved the writing of our submission in general.

  • The description of figures (especially 2 and 5) is improved.
  • A missing formula symbol for equation 5 is explained.
  • Smaller improvements across the manuscript.
  • Added missing abbreviations to the corresponding table.